# Prevalence of Obesity Among Elementary School Children in Cyprus: The National COSI Program

**DOI:** 10.3390/nu17071213

**Published:** 2025-03-30

**Authors:** Eliza Markidou, Eleftheria C. Economidou, Demetris Avraam, Maria Hassapidou, John Minas Hadjiminas, Elpidoforos S. Soteriades

**Affiliations:** 1Department of Dietetics, Ministry of Health, 1448 Nicosia, Cyprus; eliza@spidernet.com.cy; 2Department of Pediatrics, Larnaca General Hospital, 6043 Larnaca, Cyprus; eleftheria.economidou@gmail.com; 3Department of Public Health, University of Copenhagen, 1353 Copenhagen, Denmark; avraam.demetris@gmail.com; 4Department of Nutritional Sciences and Dietetics, School of Health Sciences, International Hellenic University, 57400 Thessaloniki, Greece; mnhass@gmail.com; 5Department of Pediatrics, Medical School, University of Nicosia, 2417 Nicosia, Cyprus; drjohn1@cytanet.com.cy; 6Healthcare Management Program, School of Economics and Management, Open University of Cyprus, 2220 Nicosia, Cyprus; 7Harvard School of Public Health, Department of Environmental Health, Environmental and Occupational Medicine and Epidemiology (EOME), Boston, MA 02115, USA

**Keywords:** epidemiology, obesity, children, Cyprus

## Abstract

**Background/Objectives**: A worldwide epidemic of overweight and obesity is an ongoing global health concern. This rise in overweight and obesity among children contributes to the increasing pattern of current and future physiological and psychological problems. Our study aimed at examining overweight and obesity among elementary school children in Cyprus. **Methods**: Data on children’s age, sex, place of residence (urban/rural) and weight/height status were collected using the WHO Childhood Obesity Surveillance Initiative (COSI) in Cyprus, using standardized measurements of children aged 6–9 years enrolled in the first and fourth class of elementary schools during the academic year 2021–2022. We describe the prevalence and distribution of overweight and obesity, while we also compare the findings with previous rounds of the COSI program. **Results:** A total of 1662 children were evaluated (830 boys and 832 girls, 952 from the first and 710 from the fourth class, and 1303 from urban and 369 from rural areas). The study showed 335 children with overweight (20.1%) and 275 with obesity (16.5%). Both overweight [158 (22.2%, 95% CI: 19.2–25.5%)] and obesity [149 (21.0%, 95% CI: 18.1–24.2%)] was significantly higher in children aged 8–9 years of age compared to younger children (6–7 years) (*p* < 0.0001) and was also higher in boys compared to girls (*p* = 0.0007). No difference was seen by place of residence or round of examination. **Conclusions:** Our study confirmed that both overweight and obesity remain at high levels over the past decade in both boys and girls in Cyprus.

## 1. Introduction

Childhood overweight and obesity is a major public health problem around the world [1]. According to a study on BMI over time among 128.9 million children, adolescents and adults, the prevalence of obesity increased considerably between 1975 and 2016 around the globe [2]. This rise in prevalence of overweight and obesity contributes to current and future physiological, metabolic and psychological problems affecting the cardiovascular health, the endocrine system and mental health and is associated with many other co-morbidities. Obesity during childhood is likely to continue into adulthood. Moreover, childhood obesity increases an individual’s risk for non-communicable diseases in adulthood, such as cardiovascular disease, diabetes and cancer, and increases an individual’s risk for adult morbidity and premature mortality [3,4].

As mentioned before, the rates of children and adolescents with overweight and obesity continue to climb at alarming rates worldwide and this phenomenon is also observed in European countries including Cyprus. A study reported on childhood overweight and obesity in Europe and examined changes from 2007 to 2017. The trends in the prevalence of overweight and obesity compared to the first round of COSI carried out in 2007/2008 to the latest of 2015/2017 in 11 European countries showed that the prevalence among boys and girls decreased in countries with high prevalence (Southern Europe) and remained stable or slightly increased in Northern and Eastern European countries [5].

With respect to Cyprus, Lazarou et al. reported in 2008 that almost one in two adults and at least one in four preadolescent children were found with overweight or obesity. Logistic regression analysis in both children and adults revealed some important socio-demographic predictors of obesity including factors of the built environment. Higher prevalence of overweight and obesity was observed in older adults, younger children, and in men, irrespective of age [6]. In addition, a cross-sectional study, undertaken in Cyprus in 2010, showed that the overall prevalence of overweight and obesity was higher in 2010 compared to 2000. Moreover, this study showed that the prevalence of obesity increased at a greater rate in school-aged boys and in rural areas. In addition, in rural areas, high maternal education background was associated with decreased odds for obesity [7]. Between the years 2001 and 2011, another study showed that approximately one third of adolescent boys and one quarter of adolescent girls in Cyprus were found with overweight or obesity [8]. Another study conducted in Cyprus documented increasing rates of overweight and obesity among a total of 14,090 11-year-old school-aged children who had been examined in 1997 and 2003. Females and children living in rural areas experienced the most striking increases [9]. A cross-sectional study of a representative sample of children 6–17 years of age performed from October 1999 to June 2000 showed that the prevalence of obesity in males was 10.3% and in females 9.1% using the NHANES I definition and 6.9 and 5.7% using the IOTF definition, respectively. There were an additional 16.9% of males and 13.1% of females defined as overweight with the NHANES I definition and 18.8 and 17.0% using the IOTF definition, respectively. The most significant associated factor for obesity was parental obesity status [10]. Although several studies on childhood obesity have been conducted in Cyprus, most of them were implemented almost 20 years ago and were mostly based on convenient population study samples.

Some additional studies conducted in Cyprus looked at potential associated risk factors. A study showed a bidirectional association between weight gain and appetite traits in infancy, suggesting that the effect of postnatal weight gain on obesity development is partly mediated by programming of appetite traits [11]. Another study showed that sedentary behaviors, such as watching TV, may be important predictors of children’s obesity compared to physical activity behaviors [12]. Furthermore, an investigation of the relationships between *n*-3 and *n*-6 fatty acids in subcutaneous fat and BMI as well as overweight status among 88 children from Crete and Cyprus revealed positive associations between adipose tissue arachidonic acid and BMI and overweight status. Specifically, the mean levels of arachidonic acid, dihomo-gamma-linolenic acid and docosahexaenoic acid were higher in subjects with overweight and obesity [13]. Parental obesity and high birth weight constitute factors that are significantly associated with obesity, while low birth weight being associated with undernutrition in preschool children [14].

The development of obesity is largely explained within a bio-socioecological framework, whereby biological predisposition, socioeconomic background and environmental factors interact together to promote deposition and proliferation of adipose tissue in the body. More specifically, there is a high degree of biological heterogeneity in bodyweight regulation and energy dynamics and, together with some bio-behavioral factors such as poor sleep quality, life adversity, prolonged stress and medications (iatrogenic weight gain), may contribute to dysfunction of the energy regulatory system leading to considerable weight gain. There is some evidence supporting genetics as a contributing factor, but the rise in obesity prevalence over the past few decades is mostly attributed to energy expenditure vs. energy intake and lifestyle choices [4]. Therefore, environmental and behavioral factors, such as diet (ultra-processed foods-UPFs) junk food [15,16], magnesium (Mg) deficiency [17], screen exposure, lower levels of moderate-vigorous physical activity, sleep deprivation, poor sleep quality and late bedtime, as well as lower parental education level, middle- and lower-household income, all contribute to the risk of obesity in children, adolescents and adults [3,6].

The WHO defines overweight and obesity as an abnormal or excessive fat accumulation that presents a risk to health. For epidemiological purposes and routine clinical practice, simple anthropometric measures are generally used as screening tools. Body mass index (BMI) (weight/height^2^; kg/m^2^) is used as an indirect measure of body fatness in children and adolescents and is compared with population growth references adjusted for age and sex. The WHO 2006 Growth Standard is recommended in many countries for children aged 0–5 years, and for children aged 0–2 years in the USA. For older children and adolescents, other growth references are used, including the WHO 2007 Growth Reference, recommended for children aged 5–19 years (overweight defined as BMI ≥ 1 SD and obesity as BMI ≥ 2 SD of the median for age and sex), and the United States Centers for Disease Control and Prevention (CDC) Growth Reference for those aged 2 to 20 years (overweight is >85th to <95th percentile and obesity is ≥95th percentile based on CDC growth charts) [3].

Early monitoring, diagnosis and management of obesity in the pediatric population should represent a high priority goal in national, regional and global public health programs. Identifying the reported rising in prevalence of overweight and obesity among children and understanding the negative impact that childhood obesity has on individual and population health could lead to interventions and improvement of the health of society. In this study, we aimed at describing the prevalence and distribution of overweight and obesity among elementary school-age children in Cyprus by age, sex and place of residence based on the WHO Childhood Obesity Surveillance Initiative (COSI) methodology in Cyprus using a nationally representative pediatric sample of school-age children. In addition, we aimed at comparing the results of the 6th round of COSI data with previous rounds of data collection from the same program in Cyprus.

## 2. Materials and Methods

The WHO established a surveillance initiative in 2007, the WHO Childhood Obesity Surveillance Initiative (COSI) program, to generate reliable and valid national-level data on the prevalence of overweight and obesity among primary school-aged children. The initiative established a common protocol which allowed systematic collection of data on children’s weight/height status by routine and standardized measurements of children aged 6–9 years old. Additional data on dietary intake, physical activity, sedentary behavior, family background and school environments were also collected. The target of the Surveillance Initiative was to monitor the obesity among children and create a database of weight/height and a database of dietary habits of children aged 6–9 years old that could be comparable among European countries and could show trends in the same countries over the years. COSI is now the childhood obesity surveillance initiative with the widest coverage in the world, with participation increasing from 13 Member States in 2007 to 45 Member States in 2020. In total, the first five rounds of data collection have yielded anthropometric data on over 1.3 million children [18].

Cyprus is part of the WHO Childhood Obesity Surveillance Initiative (COSI) from the first round that started in 2007. Since then, Cyprus participated in all 6 consecutive rounds. During round 4, Cyprus participated in data collection of weight/height and data regarding the school environment. The same data were also collected for rounds 5 (years 2018–2020) and 6 (years 2021–2022). The study was approved by the WHO COSI coordination team as well as by the Cyprus Ministry of Health. In addition, the study has been approved by the Cyprus National Bioethics Committee (ΕΕΒΚ ΕΠ 2024.01.229, 8 August 2024) for secondary data analyses.

### 2.1. Study Population

The study population consisted of children aged 6–9 years old enrolled in the elementary schools of Cyprus. Children in the 1st and 4th grade of Cyprus elementary schools aged 6–7 and 8–9, respectively, were selected to take part in the COSI survey. Specifically, the target population was selected, in the 4th round, from 38 schools. These schools were selected using the random sampling method according to the protocol of the WHO. The selected schools remained the same in the consecutive 5th and 6th COSI rounds as recommended by the WHO. Children of the corresponding age from all randomly selected schools were invited to participate. Participation from the selected schools was almost universal. A small number of children who were absent on the day of measurements were not included in the study.

### 2.2. Study Sample Size

In the 4th, 5th and 6th rounds of COSI, we have collected data from 2350, 2340 and 2165 children, respectively. In the 6th round, we recorded sex for 2030 children. From those, 1669 were aged between 6 and 9 years old and 6 of those were considered as outliers since they had zBMI below −5 or above +5 Z-scores. The sample size constituted about 40% of the total population of children in the elementary schools of Cyprus. The final sample size for the 6th round with complete information on age, sex, zBMI and place of residence was 1662 children (Figure 1).

### 2.3. Data Collection Tools

The COSI initiative established a common protocol which enables the systematic collection of data from children across Europe and beyond. Two data collection forms were used: children’s form (completed by the researcher) and school form (completed by the school director). The first form, called children’s form, enabled systematic collection of data on children’s date of birth (or age), sex, place of residence (village/city), breakfast consumption on the day of measurement, date of measurement and weight/height status, using routine and standardized measurements. To avoid any miscalculation, the Body Mass Index (BMI), the ratio of weight to height, was calculated in a second phase. Every child’s parent provided consent before proceeding with the measurements. It is well known that there is a tendency of underreporting of weight and over-reporting of height when the parent and/or child report such measurements [19]. Therefore, measuring the weight and height by a trained person leads to a more accurate BMI estimation. The second form, called school form, consisted of questions collecting data about the characteristics of the school environment such as the frequency of physical education classes, the types of food sold in the school canteens, the availability of drinking water and special events promoting exercise and nutrition organized in the schools.

### 2.4. Data Collection and Management

The data collection took place during the academic year 2021–2022. Two trained examiners collaborated in the process of taking anthropometric measurements. The measurements were taken twice for each child to avoid mistakes, before the first school break. The weight and height were reported as well as the child’s age in months, sex, date and time of measurement and clothes worn during the process of child examination. Children were measured wearing normal light indoor clothes, without wearing jackets or shoes. The children’s outfits were identical since all primary schools have the same school uniform. The examiners were clinical dietitians who collaborated with graduate students from a master’s degree of clinical Dietetics program and attended a one-day course in weight and height measurement using standardized procedures. They were also trained in keeping confidentiality, privacy and objectivity throughout the process. Special concern and attention were being taken regarding the sensitivity of children’s weight and body image to reduce the risk of stigmatization and bullying.

All schools were required to use the same highly accurate and precise instruments, that were provided by the WHO to the COSI examiners in Cyprus. Portable electronic digital scales were used to measure the weight calibrated to 0.1 kg and measuring up to 150 kg. The height was measured in centimeters and the readings were taken to the last completed millimeters (0.1 cm) using a portable height board (stadiometer) provided by the WHO. The body mass index (BMI), metric weight (kg)/[height (m)^2^], was calculated by the examiners after all measurement procedures were finalized. Anonymity of children was kept by using a code assigned to each child.

### 2.5. Statistical Analyses

We first calculated the age of children both in days and in years as the difference between the date of measurement and the date of birth. For each child we then calculated its BMI-for-age (BMI/A) z-scores using their weight in kg, height in cm, sex (1 for boys and 2 for girls) and age in days. The cut-offs recommended by the WHO in 2007 for school-aged children and adolescents were used to compute the BMI/A z-scores and to estimate the prevalence of overweight and obesity. For the calculations of BMI/A z-scores we used the getWGSR function (Calculate WHO Growth Reference z-score for a given anthropometric measurement) from the zscorer package (version 0.3.1) in R open statistical package. Overweight was defined by the proportion of children with BMI/A above +1 z-scores while obesity was defined by the proportion of children with BMI/A above +2 z-scores. According to WHO definitions, the prevalence estimates of children with overweight include those with obesity. Children with biologically implausible (or extreme) BMI/A values were excluded from the analysis. Implausible values were defined as BMI/A values below −5 or above +5 z-scores [20,21]. A *p*-value of less than 0.05 was accepted for statistical significance.

## 3. Results

A total of 1662 children (Figure 1) were included in the current analysis. From those, 952 (52.3%) were 6–7 years old and 710 (42.7%) were 8–9 years old, while 830 (49.9%) were males and 832 (50.1%) females. A total of 1303 (78.4%) were living in urban areas and 359 (21.6%) in rural areas (Table 1). From the total of 1662 children, we found 1052 with normal weight, 335 with overweight and 275 with obesity. In Table 1 we delineate the above statistics and demographics as distributed between children with normal weight, overweight and obesity from elementary schools in Cyprus. In the age group 6–7 years old, we found 177 children with overweight and 126 with obesity. Correspondingly, in the 8–9-year-old age group, we found 158 children with overweight and 149 with obesity (*p* < 0.05). Moreover, from the total number of 335 children with overweight, 163 (48.7%) were boys and 172 (51.3%) were girls; while, from the total number of 275 children with obesity, 166 (60.4%) were boys (*p* < 0.05). Finally, the prevalence of overweight and obesity was not statistically significantly associated with the place of residence (urban vs. rural) (*p* = 0.50).

In Figure 2, we present the distribution of overweight and obesity between boys and girls. It is evident from Figure 2 that both overweight and obesity appear to remain at high levels during all consecutive rounds fourth, fifth, and sixth, with a statistically significant difference in obesity between boys in comparison with girls.

In Figure 3, we delineated the prevalence of overweight and obesity between the two different age groups. It is demonstrated that bοth overweight and obesity appear at much higher levels in the 8–9 years old age group compared to the 6–7 years old age group. Finally, in Figure 4, we compare children with overweight and obesity from urban and rural areas and the prevalence of overweight and obesity in students aged 6–9 was not statistically significantly associated with the residence in urban and rural areas in Cyprus, according to COSI rounds four, five and six.

In Table 2, we have tabulated the comparison of children with normal weight compared to those with either overweight or obesity for both age groups. The majority of overweight and obesity is seen among children 8–9 years old and mostly among boys. The prevalence of overweight and obesity combined was not statistically significantly in association with the place of residence (urban vs. rural) (*p*-value = 0.28). It is evident from all different tabulations and comparisons that both overweight and obesity appear to remain at high levels during all consecutive rounds fourth, fifth, and sixth spanning over the past decade in both boys and girls in elementary schools in Cyprus.

## 4. Discussion

Documenting the reported rise in prevalence of overweight and obesity among children and understanding the negative impact of childhood obesity on individual and population health in instrumental in designing appropriate policies and effective interventions for the improvement of the health of the society. In our study, we have used a nationally representative sample of children aged 6–9 years old enrolled in the elementary schools of Cyprus. In the current study, we focused on the prevalence and distribution of overweight and obesity among children by age, sex and place of residence based on the implementation of the WHO Childhood Obesity Surveillance Initiative (COSI) in Cyprus. This study confirmed that both overweight and obesity appears to remain at high levels during all consecutive fourth, fifth, and sixth round of data collection, spanning over the past decade in both boys and girls.

Having results from anthropometric measurements for three consecutive COSI rounds spanning from 2015 to 2022, we have the opportunity to compare the childhood obesity rate and identify changes over time during all these years. Unfortunately, the time trend in childhood obesity in Cyprus does not show any significant decreasing pattern although there is some indication of a small decrease in the last round. As noted above, through our study we provide research findings from an official and nationally representative sample of the child population of Cyprus based on the WHO COSI surveillance program. There is no doubt that the number of children with overweight and obesity have remained at a very high level throughout the past decade. During the fourth and fifth rounds, Cypriot children had the highest overweight and obesity rates (43%) among all 38 participating countries [22,23,24]. Unfortunately, the findings of the sixth round show that childhood overweight and obesity in Cyprus continues to be at the highest level compared to the other 40+ COSI participating countries.

As we mentioned before, the prevalence of overweight and obesity was not statistically significantly associated with the place of residence (urban or rural). This result can be explained by the urban–rural typology of Cyprus. Cyprus is a small island country, and it is among the smallest countries in the world, with a population of only approximately 1 million people. The distances between the different regions in Cyprus are small, with no substantial differences in the access to various amenities, transportation facilities, entertainment and education opportunities as well as health facilities between urban and rural areas. Based on the Regulation (EC) No. 1059/2003 of the European Parliament and of the Council on the establishment of a common classification of territorial units for statistics (NUTS), Cyprus is composed of single NUTS level 3 regions. This explains why all the area of Cyprus is classified as an intermediate region altogether. According to the urban–rural 2021 typology, there are no predominantly urban and there are no predominantly rural regions for Cyprus. Intermediate regions that form part of the urban–rural typology are NUTS level 3 regions, where more than 50% and up to 80% of the population live in urban clusters [25]. Therefore, it is no surprise that we could not identify any significant differences based on the comparison of this characteristic.

The children age groups were selected because they predate puberty; therefore, they are not affected by hormonal changes [26]. Obesity during childhood is likely to continue into adulthood and is associated with all predisposing factors including behavioral and environmental and could lead to cardiometabolic and psychosocial comorbidities as well as premature mortality [3]. Childhood is also critical because it is associated with adiposity rebound. BMI increases during the first year of life and again throughout childhood, and this second rise is referred to as the adiposity rebound. It is worth mentioning that numerous studies have suggested that early adiposity rebound is a predictive marker of obesity in later childhood, adolescence and adulthood [27]. Since excess adiposity during early childhood has an influence on the process of growth and puberty, the monitoring, diagnosis and management of obesity in these pediatric age groups should be a priority target for national and international public health programs [1].

The sixth COSI round was implemented in 46 countries of which each country has collected data at least once with some countries participating in all rounds such as Cyprus. When examining the most recent data from rounds fourth, fifth and sixth from all countries, the highest prevalence of overweight and obesity both in boys and girls were found in countries of Southern Europe such as Cyprus, Greece and Italy, with Cyprus having the highest rate from all countries. Although there is a slight decrease in the sixth round, Cyprus remains the leading country with the highest obesity rate. Comparing COSI findings, we realize that there are striking differences between countries, with country specific prevalence of overweight among children aged 7–9 years ranging for 6% in Tajikistan to 43% in Cyprus (fifth round). The highest prevalence of overweight among children (both sexes combined) was observed in Cyprus, Greece, Italy and Spain. These results are consistent with the north–south gradient reported, showing that children of South Europe had the highest weight compared with children from Tajikistan, Denmark and Kazakhstan with the lowest weight. Examination of the published fifth round results for the prevalence of obesity with data from the previous fourth round (2015–2017) showed that the prevalence of obesity was higher in boys than girls in all countries except Lithuania, Portugal, Slovakia and Tajikistan [22]. For example, a comparative European study showed that among boys, the highest decrease in overweight (including obesity) was observed in Portugal (from 40.5% in 2007/2008 to 28.4 in 2015/2017) and in Greece for obesity (from 30.5% in 2009/2010 to 21.7% in 2015/2017). Lithuania recorded the strongest increase in the proportion of boys with overweight (from 24.8% to 28.5%) and obesity (from 9.4% to 12.2%). The trends were similar for boys and girls in most countries [5]. The COSI program allows us also to compare results in each country over time. In the case of Cyprus, since the fourth round, we continue to find the highest rates of overweight and obesity among all the participating countries [22,23].

We would like to acknowledge some limitations. Body mass index represents a weight measurement without differentiating the percent of fat versus muscle [28]. Measurement of waist circumference and waist–hip ratio (i.e., the waist circumference divided by the hip circumference) is also suggested as an additional measure of body fat distribution. The ratio can be measured more precisely than skin folds and it provides an index of both subcutaneous and intra-abdominal adipose tissue. Measurement of waist circumference and waist–hip ratio is important when available, since excess visceral fat (regardless of total weight) is associated with metabolic abnormalities such as dyslipidaemia or hyperinsulinemia. Furthermore, abdominal or central obesity is associated with increased cardiometabolic risk in children and adolescents and specifically is associated with an increased risk of myocardial infarction, stroke and premature death [29,30,31]. Also, the average waist circumference remains a strong indicator even at older ages. For waist circumference there are regional and international growth references allowing adjustment for age and sex [32,33]. A waist-to-height ratio of more than 0.5 is increasingly used as an indicator of abdominal adiposity in clinical and research studies, with no need for a comparison reference [3]. The field researchers were using the same instruments and followed specific uniform training before the field work was implemented and were given associated guidelines for the field work. However, inter-valuator variations were not assessed. Our study strengths include its national representative sample and its repeated measurements over time.

Careful screening and assessment by providers using growth charts is crucial to prevent obesity in children and adolescents. Effective programs on dietary education and nutrition as well as promotion of physical activity represent a recommended approach to prevent overweight and obesity [4]. A multifaceted public health policy approach is required for reversing the current obesity epidemic in Cyprus. There is evidence in the international literature indicating that no single policy option appears to be unique in combating obesity, but rather there is a need to combine complementary actions on the individual and population level. Specifically, measures are needed to improve levels of knowledge and understanding regarding food, diet, health and fitness beginning from early childhood with educators and health professionals having an important role in this regard. These measures should be coupled with appropriate population campaigns emphasizing the improvement of nutritional information labeling system, and the control of food and drink advertising. There is also a consensus regarding the need for modifying the supply of healthy foods in school canteens, as well as environmental interventions at neighborhood level. Practical feasibility, social acceptability, effectiveness and social benefits of the different options and recommendations are deemed the most important criteria for a successful multifaceted policy intervention [34].

## 5. Conclusions

Although obesity has many etiological factors—biological, environmental, behavioral and individual—over the past few decades, the rise in obesity prevalence has been influenced by changes in the broader obesogenic environment [3]. Schools can be ideal sites for interventions given that children and adolescents in most parts of the world spend a substantial amount of time attending school [35]. Childcare and school environments have a direct impact on children’s nutrition through food available in the school canteen, through nutrition education within the school and the exclusion of the promotion and consumption of energy dense, micronutrient-poor foods and a high intake of sugar sweetened beverages [3]. Moreover, schools play a critical role in increasing physical activity and improving fitness among children and adolescents and improving their body composition [35]. An important public health action could also include the establishment of a national institute of nutrition and the development of a national strategic plan delineating comprehensive multidisciplinary interventions to control the current childhood obesity epidemic. Literature from European COSI programs show heterogeneity in the prevalence between countries indicating that there are opportunities for national policymakers to learn from experiences across Europe and adopt what seems to be working. Our results can mobilize countries to prioritize actions in addressing specific important parameters including physical activity behaviors that may be more effective in increasing overall exercise levels and reducing sedentary behavior [36]. In summary, our study findings call for an urgent development of a national strategic plan to combat this national epidemic of overweight and obesity among elementary school-age children in Cyprus and also serve as a highlighted example for Europe.

## Figures and Tables

**Figure 1 nutrients-17-01213-f001:**
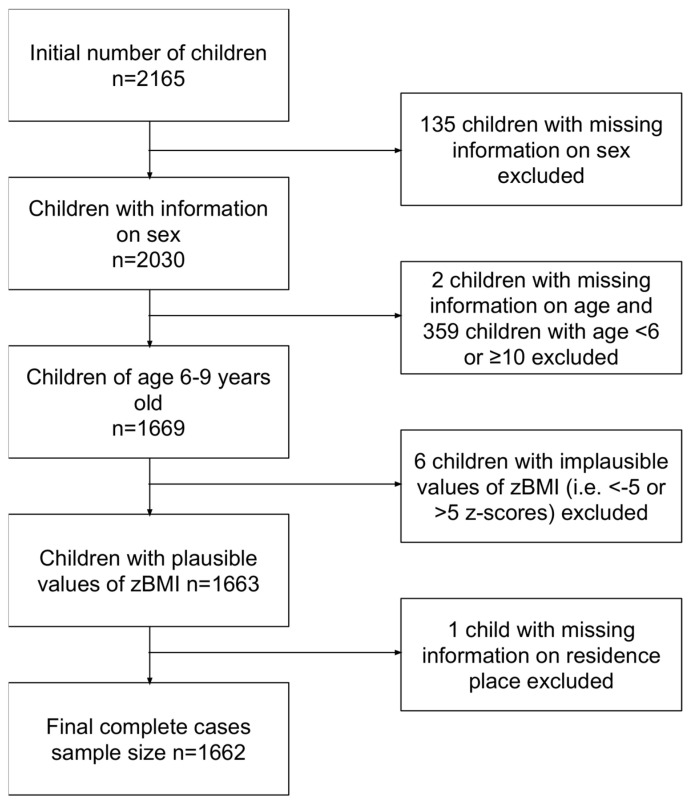
Flow chart showing the selection of the study sample with final statistical analysis for data of COSI from round 6.

**Figure 2 nutrients-17-01213-f002:**
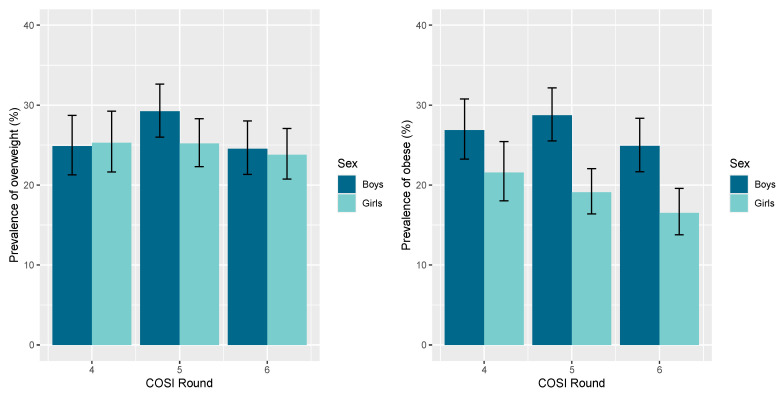
Prevalence of overweight (**left**) and obesity (**right**) in boys and girls aged 6–9 years in Cyprus, according to COSI rounds 4th, 5th and 6th. The error bars indicate the 95% confidence intervals for all proportions and are estimated by binomial exact calculations.

**Figure 3 nutrients-17-01213-f003:**
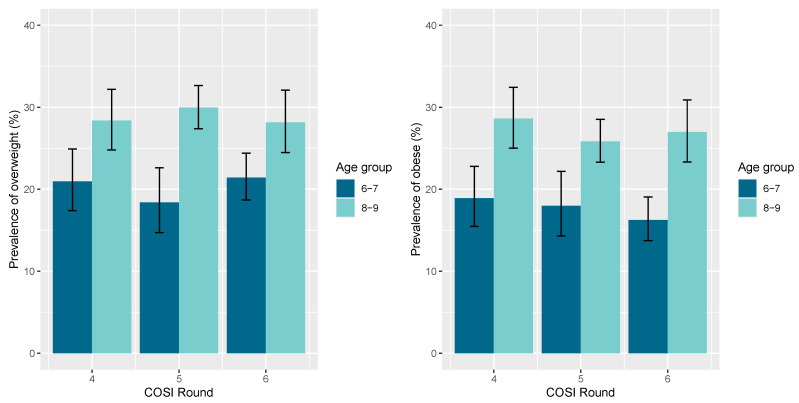
Prevalence of overweight (**left**) and obesity (**right**) in children aged 6–9 years in Cyprus, according to COSI rounds 4th, 5th and 6th. The error bars indicate the 95% confidence intervals for all proportions and are estimated by binomial exact calculations.

**Figure 4 nutrients-17-01213-f004:**
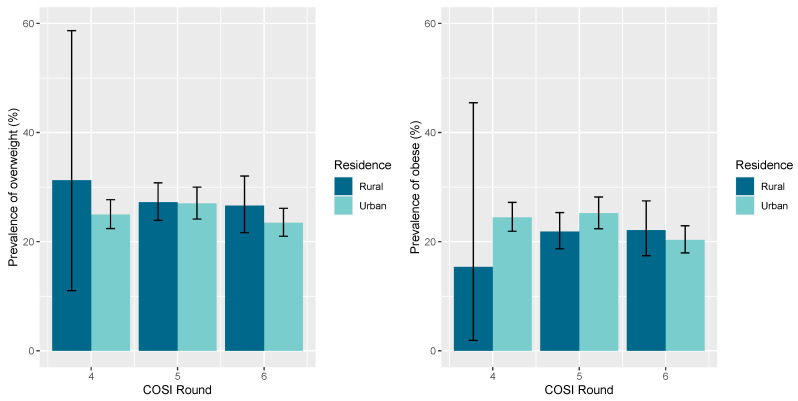
Prevalence (%) of overweight (**left**) and obesity (**right**) in students aged 6–9 years of age with residence in urban and rural areas in Cyprus, according to COSI rounds 4th, 5th and 6th. Please note that the percentages of overweight and obese living in rural areas for COSI round 4 were calculated on the available data of only 18 and 13 students, respectively. Rural areas in Cyprus include also semi-urban areas. The error bars indicate the 95% confidence intervals for all proportions and are estimated by binomial exact calculations.

**Table 1 nutrients-17-01213-t001:** Frequence, prevalence and 95% confidence interval (CI) of elementary school children with normal weight, overweight and obesity in Cyprus by age, sex and place of residence in 2021–2022 COSI survey.

	Total (n = 1662)	Normal (n = 1052)	Overweight (n = 335)	Obesity (n = 275)	*p*-Value *
Age group					<0.0001
6–7 years old	952 (57.3%, 95%CI: 54.9–59.7%)	649 (68.2%, 95%CI: 65.1–71.1%)	177 (18.6%, 95%CI: 16.2–21.2%)	126 (13.2%, 95%CI: 11.2–15.6%)	
8–9 years old	710 (42.7%, 95%CI: 40.3–45.1%)	403 (56.8%, 95%CI: 53.0–60.4%)	158 (22.2%, 95%CI: 19.2–25.5%)	149 (21.0%, 95%CI: 18.1–24.2%)	
Sex					0.0007
Boys	830 (49.9%, 95%CI: 47.5–52.4%)	501 (60.4%, 95%CI: 56.9–63.7%)	163 (19.6%%, 95%CI: 17.0–22.5%)	166 (20.0%%, 95%CI: 17.3–22.9%)	
Girls	832 (50.1%, 95%CI: 47.6–52.5%)	551 (66.2%, 95%CI: 62.9–69.4%)	172 (20.7%%, 95%CI: 18.0–23.6%)	109 (13.1%%, 95%CI: 10.9–15.6%)	
Place of Residence:					0.4964
Urban	1303 (78.4%, 95%CI: 76.3–80.4%)	834 (64.0%%, 95%CI: 61.3–66.6%)	256 (19.6%%, 95%CI: 17.5–21.9%)	213 (16.4%%, 95%CI: 14.4–18.5%)	
Rural	359 (21.6%, 95%CI: 19.6–23.7%)	218 (60.7%%, 95%CI: 55.5–65.8%)	79 (22.0%%, 95%CI: 17.8–26.7%)	62 (17.3%%, 95%CI: 13.5–21.6%)	

* *p*-value of the Pearson’s Chi-squared tests between each demographic variable and the categorical zBMI variable assessing the null hypothesis that the two variables are independent. The null hypothesis is rejected for *p*-value < 0.05. Notes: (i) Rural areas include also semi-urban areas. (ii) The 95% confidence intervals for all proportions are estimated by binomial exact calculations. (iii) The percentages in the column of total sample sum up to 100% for each demographic variable, while the percentages in the columns of normal, overweight and obesity sum up to 100% for each category of each demographic variable.

**Table 2 nutrients-17-01213-t002:** Comparison between children with normal weight vs. children with overweight and obesity in Cyprus by age, sex and place of residence in 2021–2022 COSI survey.

	Normal (n = 1052)	Overweight and Obesity (n = 610)	*p*-Value *
Age group			<0.0001
6–7 years old (n = 952)	649 (68.2%, 95%CI: 65.1–71.1%)	303 (31.8%, 95%CI: 28.9–34.9%)	
8–9 years old (n = 710)	403 (56.8%, 95%CI: 53.0–60.4%)	307 (43.2%, 95%CI: 39.6–47.0%)	
Sex			0.0151
Boys (n = 830)	501 (60.4%, 95%CI: 56.9–63.7%)	329 (39.6%, 95%CI: 36.3–43.1%)	
Girls (n = 832)	551 (66.2%, 95%CI: 62.9–69.4%)	281 (33.8%, 95%CI: 30.6–37.1%)	
Place of Residence			0.2799
Urban (n = 1303)	834 (64.0%, 95%CI: 61.3–66.6%)	469 (36.0%, 95%CI: 33.4–38.7%)	
Rural (n = 359	218 (60.7%, 95%CI: 55.5–65.8%)	141 (39.3%, 95%CI: 34.2–44.5%)	

* *p*-value of the Pearson’s Chi-squared tests between each demographic variable and the categorical zBMI variable assessing the null hypothesis that the two variables are independent. The null hypothesis is rejected for *p*-value < 0.05. Notes: (i) Rural areas include also semi-urban areas. (ii) The 95% confidence intervals for all proportions are estimated by binomial exact calculations. (iii) The percentages sum up to 100% for each category of each demography variable.

## Data Availability

Data are available upon formal request.

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
