# Peer review of "Prevalence of Obesity Among Elementary School Children in Cyprus: The National COSI Program"

_nutrients, 2025, doi:10.3390/nu17071213_

Round 1

Reviewer 1 Report

Comments and Suggestions for Authors

This is a survey to estimate overweight and obesity in children aged 6 to 9 years old attending elementary schools in Cyprus. Information was collected on age, sex, place of residence (urban/rural) and weight / height status, by standardized measurements of children aged 6 – 9 years enrolled in the 1st and 4th class of elementary schools of Cyprus during the academic year 2021 – 2022. A total of 1 662 children (830 boys and 832 girls, 952 from the 1st and 710 from the 4th class, 1 303 from urban and 369 from rural areas). The study showed 335 children with overweight (20.1%) and 275 with obesity (16.5%). Both overweight and obesity were significantly higher in the children aged 8 – 9 years of age compared to the younger children (6 – 7 years) and was also higher in boys compared to girls. No difference was seen by place of residence or round of examination.

The study is well described and presents clear results and discussions. I am sending suggestions that could improve it.

Introduction
- Modify the order of the first three paragraphs and adapt them so that the presentation of the estimates of overweight and obesity in children from Cyprus is immediately after the global estimates.

Methods
- The manuscript informs that the anthropometric measurements were performed by two trained evaluators. Was any protocol followed for the anthropometric assessment? Were the accuracy and inter-evaluator variations measured?

Results
- The quality/definition of Figure 1 (flow chart) was compromised. If possible, insert the image in higher quality.

- In the tables, consider showing the exact value of the p-value to three decimal places. Avoid <0.05.

- It would be important to present the 95% CI for all the estimated prevalences of overweight and obesity (Table 1 and 2 Figures 2, 3 and 4).

Discussion
- It would be important to further discuss the differences observed in the prevalence of outcomes in Cyprus compared to other regions.
- Consider other limitations of the study related to the method, for example, a possible inter-evaluator variation in anthropometric measurement, if this was not assessed.

Reviewer 2 Report

Comments and Suggestions for Authors

General Comments

This study investigates the prevalence of obesity and overweight among elementary school children in Cyprus based on data from the WHO Childhood Obesity Surveillance Initiative (COSI). The research is well-structured, presents a thorough analysis of obesity trends, and provides significant insights into public health concerns. However, there are some aspects that require further clarification, methodological refinement, and improved discussion of findings.

Section-Specific Review

Abstract

Strengths:

  • The abstract succinctly outlines the study’s objectives, methodology, key results, and conclusions.
  • The prevalence rates are clearly stated, and the comparison with previous data rounds is informative.

Areas for Improvement:

  • Consider including more details on the statistical significance of findings.
  • Clarify the impact of the findings on public health interventions.

Introduction

Strengths:

  • The introduction effectively contextualizes childhood obesity as a global issue.
  • The relevance of the study within Cyprus and its comparison to other countries enhances its significance.

Areas for Improvement:

  • The introduction should more explicitly state the research gap that this study aims to fill.
  • Additional references to recent studies in similar European contexts could provide a stronger background.

Materials and Methods

Strengths:

  • The methodology is clearly defined, with a thorough description of the study population, data collection, and statistical analysis.
  • The use of standardized WHO protocols ensures data reliability.

Areas for Improvement:

  • Provide more justification for the sample size calculation and its statistical power.
  • Discuss potential biases, such as self-selection bias or any limitations in data collection methods.
  • Elaborate on the inclusion and exclusion criteria for participant selection.

Results

Strengths:

  • The results are well-structured, with clear tables and figures illustrating obesity prevalence by age, sex, and place of residence.
  • The statistical analyses, including p-values, provide strong support for the findings.

Areas for Improvement:

  • Some results could benefit from additional context, particularly in terms of their practical significance.
  • Further statistical tests could strengthen conclusions, such as regression analyses to identify risk factors.
  • Address any unexpected findings and their potential explanations.

Discussion

Strengths:

  • The discussion effectively places the findings within a broader public health context.
  • The comparison with previous COSI rounds provides useful insights into trends over time.

Areas for Improvement:

  • The discussion should further elaborate on the policy implications of the findings.
  • Consider adding more details on intervention strategies that could address obesity prevalence in Cyprus.
  • The limitations section should discuss possible measurement errors, potential confounding variables, and generalizability concerns.

Conclusion

Strengths:

  • The conclusion summarizes key findings concisely and effectively.
  • The call for urgent intervention aligns well with the study's public health relevance.

Areas for Improvement:

  • Consider suggesting specific public health initiatives or policies based on the results.
  • Ensure that the conclusions are fully supported by the data presented.

Final Recommendation

  • Accept with Minor Revisions: The manuscript presents a well-executed study with valuable findings. Minor revisions are required to enhance clarity, methodological transparency, and discussion depth.

Suggested Revisions Summary

  1. Strengthen the abstract by including statistical significance details.
  2. Clearly define the research gap in the introduction.
  3. Justify sample size selection and discuss potential biases in the methodology.
  4. Expand statistical analyses in the results, considering additional tests.
  5. Deepen the discussion on policy implications and intervention strategies.
  6. Elaborate on limitations, including potential confounding variables.
  7. Provide specific recommendations for public health actions in the conclusion.

Once these revisions are addressed, the manuscript will significantly contribute to the field of childhood obesity research and public health policy.
